# Advances and Challenges in Sepsis Management: Modern Tools and Future Directions

**DOI:** 10.3390/cells13050439

**Published:** 2024-03-02

**Authors:** Elena Santacroce, Miriam D’Angerio, Alin Liviu Ciobanu, Linda Masini, Domenico Lo Tartaro, Irene Coloretti, Stefano Busani, Ignacio Rubio, Marianna Meschiari, Erica Franceschini, Cristina Mussini, Massimo Girardis, Lara Gibellini, Andrea Cossarizza, Sara De Biasi

**Affiliations:** 1Department of Medical and Surgical Sciences for Children & Adults, University of Modena and Reggio Emilia, 41125 Modena, Italy; elena.santacroce@unimore.it (E.S.); miriam.dangerio@unimore.it (M.D.); alinliviu.ciobanu@unimore.it (A.L.C.); linda.masini@studenti.unipr.it (L.M.); domenico.lotartaro@unimore.it (D.L.T.); lara.gibellini@unimore.it (L.G.); andrea.cossarizza@unimore.it (A.C.); 2Department of Surgery, Medicine, Dentistry and Morphological Sciences, University of Modena and Reggio Emilia, 41121 Modena, Italy; irenecoloretti@gmail.com (I.C.); stefano.busani@unimore.it (S.B.); mariannameschiari1209@gmail.com (M.M.); erica.franceschini@unimore.it (E.F.); cristina.mussini@unimore.it (C.M.); massimo.girardis@unimore.it (M.G.); 3Department of Anesthesiology and Intensive Care Medicine, Center for Sepsis Control and Care, Jena University Hospital, 07747 Jena, Germany; ignacio.rubio@med.uni-jena.de

**Keywords:** sepsis, immune response, lymphopenia, phenotyping, biomarkers, flow cytometry, personalized medicine

## Abstract

Sepsis, a critical condition marked by systemic inflammation, profoundly impacts both innate and adaptive immunity, often resulting in lymphopenia. This immune alteration can spare regulatory T cells (Tregs) but significantly affects other lymphocyte subsets, leading to diminished effector functions, altered cytokine profiles, and metabolic changes. The complexity of sepsis stems not only from its pathophysiology but also from the heterogeneity of patient responses, posing significant challenges in developing universally effective therapies. This review emphasizes the importance of phenotyping in sepsis to enhance patient-specific diagnostic and therapeutic strategies. Phenotyping immune cells, which categorizes patients based on clinical and immunological characteristics, is pivotal for tailoring treatment approaches. Flow cytometry emerges as a crucial tool in this endeavor, offering rapid, low cost and detailed analysis of immune cell populations and their functional states. Indeed, this technology facilitates the understanding of immune dysfunctions in sepsis and contributes to the identification of novel biomarkers. Our review underscores the potential of integrating flow cytometry with omics data, machine learning and clinical observations to refine sepsis management, highlighting the shift towards personalized medicine in critical care. This approach could lead to more precise interventions, improving outcomes in this heterogeneously affected patient population.

## 1. Introduction

As critical care medicine continues to evolve, significant updates in the understanding and management of sepsis have emerged. The 2023 guidelines have notably redefined sepsis as “life-threatening organ dysfunction resulting from a dysregulated host response to infection”, a pivotal shift that has reformed the clinical and conceptual approach to this condition [1,2]. This redefinition emphasizes the acute severity and potential lethality of sepsis, highlighting the urgency for rapid diagnosis and effective therapeutic intervention. Further underscoring its significance, the World Health Organization has recognized sepsis as a global health priority [3].

The current treatment regimen, primarily consisting of antibiotics, fluid therapy, and organ support, has yet to significantly improve patient outcomes, contributing to reduce the persistently high mortality rates, which range from 38.6% to 80%, depending on the patient population and study [4]. Despite advances in treatment, sepsis remains a formidable challenge in clinical care, remaining a leading cause of mortality worldwide, as evidenced by high in-hospital mortality rates. It is estimated to account for about 20% of all global mortality, making it one of the most life-threatening conditions encountered in emergency departments [5,6,7]. In 2017 alone, sepsis accounted for nearly 49 million cases and 11 million deaths, reflecting its widespread impact [5,8,9].

Systemic inflammation is a characteristic feature of sepsis. Clinical symptoms include changes in body temperature (fever or hypothermia), leukocytosis and leukopenia, tachycardia, hypotension, and hyperventilation. It is possible, however, that they may also originate from non-infectious sources such as trauma or so-called sterile inflammation and are neither specific nor sensitive in detecting sepsis. Consequently, it is necessary to use markers that, by enabling early detection of sepsis and organ dysfunction, allow for early and targeted interventions. C-reactive protein (CRP) is a sensitive parameter for diagnosing non-systemic infections, whereas procalcitonin (PCT) appears to be a useful parameter for improving the diagnosis and monitoring of therapy in patients with sepsis and septic shock [10]. Also, tools like the Sequential sepsis-related Organ Failure Assessment (SOFA) score have been instrumental in assessing the severity of organ dysfunction and predicting mortality risk, aiding in early diagnosis and management [2,11].

A key challenge in sepsis management is the heterogeneity of patient populations, encompassing diverse underlying conditions and varying immune responses. This diversity complicates the development of universally effective treatment strategies [12]. This complexity is further compounded by recent research which brings to light the nuanced interplay between pre-sepsis host factors, the evolution of lymphopenia, and persistent changes in immune function following sepsis recovery. Such insights reveal the intricate and dynamic nature of sepsis immunology, underlining the importance of a more detailed and personalized approach to its understanding and management. This shift in perspective, advocating for comprehensive immunological profiling and considering the microbiome’s influence, aims to refine our strategies in sepsis prevention, diagnosis, and treatment [13].

The rising trend in sepsis cases is attributed to factors such as an aging population, increased use of invasive surgical procedures, widespread use of immunosuppressive medications and chemotherapy, and the escalating challenge of antibiotic resistance [14]. Particularly at risk are individuals with compromised immune systems, such as those with HIV/AIDS, cirrhosis, or those undergoing cancer treatments. A French ICU study indicated a nearly threefold (Odds Ratio 2.8) increased chance of sepsis in these groups [15].

Approximately a third of sepsis or septic shock patients described in a 1997–2011 study was overtly immunocompromised [16]. Cancer patients, in particular, face a quadrupled risk of sepsis and higher mortality rates [17]. Genetic predispositions also contribute to sepsis susceptibility. A Danish study highlighted an increased risk of infection-related death before age 50 if a biological parent had died of an infectious cause [18]. Lifestyle factors, such as alcohol consumption and smoking, have been associated with increased sepsis risk and complications [19]. Intriguingly, vitamin D deficiency has been linked to a higher sepsis risk, though the effectiveness of supplementation is unclear [20].

Notably, systemic inflammation is not always caused by infections. Conditions like pancreatitis, trauma, or severe allergic reactions can mimic septic states. The EPIC II study provided valuable insights into the infectious causes of sepsis, revealing the lung as the most common infection site among patients from 75 countries [21]. While these statistics and trends underscore the clinical urgency and complexity of sepsis management, they also point towards the necessity of a deeper understanding of its underlying mechanisms. Sepsis, far from being a singular pathologic entity, manifests through diverse physiological disruptions at various levels of human biology. This intricate web of pathophysiological changes, stemming from and extending beyond the infection site, necessitates a thorough exploration of its pathogenesis and cellular mechanisms. Such an understanding is crucial not only for enhancing diagnostic accuracy and treatment efficacy but also for paving the way towards personalized medicine in sepsis care.

## 2. Sepsis Pathophysiology: A Multi-Level Perspective

Sepsis presents a unique challenge due to its multi-level pathogenesis involving molecular, cellular, and organ systems. Understanding these layers is crucial for developing new effective diagnostic and therapeutic strategies.

### 2.1. Molecular and Immune Mechanisms in Sepsis

The pathogenesis of sepsis extends beyond the infection type, encompassing a spectrum of biological processes, including inflammation, coagulation, endothelial activation, and alterations in the microbiome [22,23].

Central to the pathogenesis of sepsis is an intricate interplay between the immune system and the relevant pathogens, ranging from bacteria to viruses and fungi. The immune response, initially protective, can become unbalanced due to the persistent stimulation of pattern recognition receptors (PRRs) like Toll-like receptors (TLRs), nucleotide-binding oligomerization domain-like receptors (NLRs), and C-type lectin receptors (CLRs). These receptors detect pathogen-associated molecular patterns (PAMPs) and damage-associated molecular patterns (DAMPs), ideally leading to pathogen elimination and restoration of homeostasis. However, in sepsis, this balanced response is often disrupted, leading to excessive inflammation and tissue injury, a state termed “hyperinflammation” [23,24,25]. The gut microbiome, vital in maintaining immune homeostasis and protecting against pathogenic invasion, undergoes significant disruption in sepsis. Dysbiosis in sepsis patients is marked by decreased microbial diversity and overgrowth of opportunistic pathogens such as Enterobacter, Enterococcus, and Staphylococcus. This dysbiosis contributes to increased gut permeability and systemic infection, affecting distant organ functions and systemic immunity [23,26,27].

In clinical settings, sepsis patients often exhibit concurrent hyperinflammation and immune suppression, resulting in variable different clinical outcomes, including persistent inflammation, immunosuppression, and catabolism syndrome (PICS) in critically ill patients. This dichotomous response highlights the complexity and variability in sepsis pathogenesis, necessitating personalized therapeutic approaches [28,29]. As shown in Figure 1 (adapted from Hotchkiss et al. [30]), the proinflammatory phase in sepsis, i.e., the hyper-inflammatory response, often termed the “cytokine storm”, is characterized by intense systemic release of cytokines and other inflammatory mediators, leading to tissue damage and organ dysfunction. This phase aligns with the concept of Systemic Inflammatory Response Syndrome (SIRS) but also acknowledges the potential harm of unchecked inflammatory responses. Neutrophils play a vital role in this phase, contributing to hyperinflammation through the release of proteases and reactive oxygen species and forming neutrophil extracellular traps (NETs). While NETs are essential for antibacterial defense, excessive NETosis can lead to tissue damage, thrombosis, and organ failure [31,32].

Inflammasomes, high molecular weight protein complexes central to the immune system, play a crucial role in sepsis. These machineries activate inflammatory caspases and cytokines of the IL-1 family. Their activation involves various components, such as plasma membrane receptors (e.g., P2 × 7R, panx-1), cytoplasmic mediators (ROS, TXNIP), and intrinsic elements like NLRPs and AIM2. The detection of PAMPs and DAMPs triggers these inflammasomes, initiating a critical immune response against sepsis [33]. The regulation of inflammasome activity is vital for modulating the immune response in sepsis. Novel therapeutic strategies focus on mitigating overactivation of inflammasomes. This includes blocking specific receptors and inhibiting mediators involved in their activation, as well as targeting the cytokines they process, such as IL-1β and IL-18 [34].

The anti-inflammatory phase, i.e., the hypo-inflammatory response (Figure 1), also referred to as “Immuno-paralysis”, is marked by reduced immune responsiveness, leaving patients susceptible to secondary infections, and complicating the recovery process. This phase involves the suppression of various immune cells, including T cells and B cells, their exhaustion, and their reprogramming through epigenetic changes, contributing to the susceptibility to secondary infections and viral reactivation [30,35]. Monocytes and macrophages in sepsis frequently exhibit a “tolerant” state, with diminished proinflammatory cytokine production and altered gene expression due to epigenetic modifications [36,37]. This state contributes to the immune suppressive environment observed in sepsis patients.

The central nervous system (CNS) also plays a role in sepsis, particularly through the neuro-immune inflammatory reflex. However, the contribution of disrupted neuro-immune interactions in sepsis-induced immune suppression remains to be fully understood [38]. The intricacy of sepsis, coupled with individual variability in immune responses influenced by health status, genetic predisposition, and infecting organisms, highlights the need for personalized therapeutic approaches in managing sepsis [30,39].

### 2.2. Sepsis Pathophysiology across Organ Systems

Sepsis pathophysiology extends to various organ systems, each affected differently:Central to this process is the cardiovascular system, which undergoes significant changes during the progression of sepsis from localized infection to severe systemic inflammation and septic shock. Despite normal or increased cardiac output, patients with sepsis often experience acute biventricular dysfunction and elevated lactate levels, indicating a critical imbalance in tissue oxygenation and metabolic dysfunction [40,41,42,43,44].At the endothelial level, sepsis induces profound alterations, such as increased leukocyte adhesion, a shift to a procoagulant state, and compromised barrier function, leading to tissue oedema and microvascular disturbances [19,45,46]. These changes, collectively described as endothelial dysfunction, coupled with widespread tissue factor expression and impaired anticoagulant mechanisms, can culminate in disseminated intravascular coagulation (DIC), further exacerbating organ dysfunction, and increasing mortality risk [47].In the liver, sepsis impairs crucial functions, including the clearance of bilirubin and processing of pathogen lipids, which intensifies systemic inflammation [48]. Septic acute kidney injury (AKI) involves cytokine and immune-mediated microvascular and tubular dysfunction, rather than mere hypoperfusion or tubular necrosis [49,50,51,52,53].

This brief overview of sepsis pathophysiology, illustrating its extensive impact on various organ systems, underscores the necessity for tailored therapeutic strategies. As we move forward, the focus shifts towards individualized care, recognizing the unique ways in which sepsis manifests in each patient. This understanding forms the basis for the next crucial step in sepsis management: the development of phenotyping and personalized approaches in treatment.

## 3. Biomarkers in Sepsis

The complexity of sepsis and difficulty of its therapy require innovative approaches, particularly considering the limitations of current treatments. Regardless of the use of broad-spectrum antibiotics, patients often struggle to clear primary infections and are susceptible to secondary infections during hospitalization. Enhancing immune competence could be pivotal in resolving primary infections and preventing fatal secondary complications [54].

Recent therapies and treatment protocols have led to prolonged illnesses with a prevalence of the immunosuppressive phase in sepsis. This condition is increasingly affecting the elderly, with a significant percentage of sepsis cases and related deaths occurring in individuals over 65 years of age in advanced healthcare systems. The diminished efficiency of the immune system in the elderly, a phenomenon known as immunosenescence, along with comorbidities, heightens the risk of sepsis incidence and mortality [55,56].

The immune response in sepsis varies and depends on individual variables such as age, cytokine profiles, immune competence, and comorbidities. Understanding the patient’s phenotype is crucial in developing a tailored therapeutic approach, considering both the disease phase and individual patient factors.

The diversity of sepsis complicates the identification of high-risk patients, early diagnosis, and disease-specific treatments. Delay in appropriate treatment, especially in administering potent antibiotic regimens, substantially increases mortality risk [57] (p. 202).

Flow cytometry, a powerful technology for detecting and monitoring multiple markers at the cellular level, is effective for assessing cell heterogeneity (Figure 2) and is a quicker, less expensive method for studying immune cells. This technique, known as immunophenotyping, is pivotal in understanding the roles of innate and adaptive immune cells in sepsis [54,58].

### 3.1. Innate Immunity

#### 3.1.1. Neutrophils

Neutrophils are the most abundant innate cell type and the first responders to sites of microbial infection [59]. They play a crucial role in the eradication of microbes as well as the survival of patients suffering from sepsis [60].

Delayed neutrophil apoptosis is one of the most prominent innate immune changes in sepsis, compared to physiological conditions in which neutrophils are constitutively pro-apoptotic. This protracted apoptotic process leads to an accumulation of neutrophils in circulation, which are of varying maturity, and to persistent neutrophil dysfunction [61,62]. In addition, the bone marrow produces and releases immature neutrophils, which can worsen delayed cell death [63]. Immature neutrophils are identified by low expression of CD10 and CD16 (CD10^low^CD16^low^) as well as their immunosuppressive functions and their presence correlates with early mortality after sepsis [64,65].

Furthermore, pro-inflammatory mediators can induce the activation of extracellular signal-regulated kinase (ERK) 2 and phosphoinositide 3-Kinase (PI3K) pathways which eventually lead to upregulation of anti-apoptotic Bcl-xL protein and downregulation of Bim [66,67]. Also, activation of Akt pathway results in phosphorylation of Bad which prevents apoptosome formation and therefore inhibits neutrophil apoptosis [68]. 

Moreover, neutrophils in sepsis display deficits in oxidative burst, with a reduced production of ROS, in cellular migration patterns, complement activation ability, and in bacterial clearance, along with decreased recruitment to infected tissues [59,69,70]. This persistent neutrophil dysfunction has been implicated in the development of hospital-acquired infections in several human studies [71]. In addition, patients with the most pronounced neutrophil dysfunction after sepsis are the most likely to develop ICU complications such as ventilator-associated pneumonia and other nosocomial infections [72].

Despite typically producing low cytokine amount, neutrophils in sepsis are notable for increased interleukin (IL)-10 production [73]. Also, apoptosis-induced lymphopenia may be caused by neutrophils secreting excessive amounts of IL-10 during sepsis, which hinders T lymphocyte proliferation [73].

Neutrophils can attack a wide range of microorganisms by forming NETs [31]. Studies suggest that NETs can prevent the spread of bacteria, especially during the early phase of infection [74,75]. The release of NETs is crucial in the pathogenesis of diseases such as sepsis, atherosclerosis, and autoimmunity [76,77]. Septic patients exhibited higher levels of NETs compared to healthy controls which have been linked to sepsis severity and organ dysfunction [32]. In sepsis and similarly to neutrophils in COVID-19, which lead to chronic basal inflammation, the system may be defined as refractory, resulting in enhanced glycolysis and glycogenolysis, which may explain why NETosis activity may have increased [78]. Additionally, NETs have been suggested to contribute to coagulation impairment in sepsis [79,80]. In vitro studies have shown that NETs promote fibrin formation and deposition, and they colocalize with fibrin in blood clots [80,81,82].

Neutrophil CD64 (nCD64) is considered a reliable and effective marker for identifying systemic infection, sepsis and tissue injury with high sensitivity and specificity [83]. nCD64 serves as a valid marker for detecting an early step immune response to bacterial infection, typically characterized by a >10-fold, rapid increase in its expression upon activation by pro-inflammatory cytokines [84,85]. As a diagnostic and prognostic biomarker, nCD64 is associated with ICU survival, mortality, and impending clinical deterioration. It could be used to differentiate sepsis severity stages and to guide antibiotic usage. Early increased percentages of CD64^+^ neutrophils and CD16^+^ monocytes are thus associated with diagnosis of sepsis [86]. The amount of plasma membrane expression of CD64, measured by flow cytometry evaluating its mean fluorescence index (MFI), can be used as a neonatal sepsis biomarker [87] and, indeed, it demonstrated a high sensitivity and specificity as a specific biomarker for systemic sepsis in adults [88]. Furthermore, its expression remains stable in EDTA-anticoagulated blood for at least 24 h at room temperature, facilitating its quantification for clinical applications [89].

Also, in septic patients, neutrophils showing elevated programmed death ligand (PD-L) 1 (a ligand that binds to the inhibitory co-receptor PD1 found on T cells) expression can induce apoptosis in CD4^+^ T lymphocytes, contributing to lymphocyte depletion, except for regulatory T cells (Tregs) [58]. Finally, it has been proposed that a subset of CD16^hi^CD62^low^ immune suppressive neutrophils possess the ability to hinder T lymphocyte proliferation [90], although possessing poor ability to kill bacteria like *Staphylococcus aureus* [91].

#### 3.1.2. Monocytes

Simply speaking, monocytes can be categorized in three main subpopulations: classical (CD14^+^CD16^−^), intermediate (CD14^+^CD16^+^), and non-classical (CD14^low^CD16^+^) monocytes. Different monocyte subsets are expanded or reduced in specific pathologies, and indeed, CD16^+^ monocytes represent up to 50% of all circulating monocytes during sepsis [92]. An important rise in both the intermediate and non-classical subsets was noted among patients with sepsis. Interestingly, after treatment, non-classical inflammatory monocytes returned to normal levels [93]. This suggests that the expansion of CD16^+^ monocytes could be used to identify the inflammatory condition.

Monocytes in sepsis exhibit significant alterations in the expression of other surface markers. Notably, the reduced expression of the HLA-DR correlates with a decreased capacity to mount a pro-inflammatory response and to present antigens effectively. Furthermore, in early deceased patients the expression of co-stimulatory molecule CD86 was decreased, while in intermediate and non-classical monocytes its expression was lowered when compared to that of those who survived [94].

The trend of HLA-DR surface expression over time, studied by Bodinier et al. [95], suggests distinct phenotypes with varied clinical outcomes with the change in HLA-DR expression as a putative prognostic marker of septic shock in ICU [96,97]. Reduced HLA-DR expression on monocytes was associated with poor outcomes in sepsis [98,99,100]. HLA-DR expression levels can be measured using flow cytometry, either as the percentage of CD14^+^ cells positive for the marker or as the mean fluorescence intensity on total monocytes. Also, the number of HLA-DR molecules can be determined with flow cytometry using specific fluorophore-labelled beads. In septic patients, an increased programmed death ligand (PD-L)1 expression [101,102,103] was found that was associated with HLA-DR downregulation [101].

Interestingly, an increased expression of CD64, a marker related to phagocytosis, was detected in monocytes from septic patients; indeed, a preserved phagocytic activity has been found in monocytes during sepsis [104] which indicates that monocytes are not anergic. Further, those who survived sepsis showed an enhanced expression of CD64 and higher levels of HLA-DR than non survivors [105].

In sepsis, monocytes show impaired functionality with a diminished capacity to release pro-inflammatory cytokines like tumor necrosis factor (TNF), IL-6, IL-12, and IL-1α. Conversely, the release of anti-inflammatory mediators like IL-10 and IL-1RA is enhanced or remains unimpaired [106,107]. Moreover, ROS and nitric oxide (NO) production by monocytes is enhanced in septic patients, at least in the first phase of sepsis (either during sepsis or septic shock) [104,108,109,110,111]. Cytokines like granulocyte-macrophage colony-stimulating factor (GM-CSF), which modulates monocyte function by upregulating HLA-DR expression, showed promise in small clinical trials, suggesting potential for sepsis therapy [54]. However, it should be noted that HLA-DR downregulation correlates with reduced TNF, IL-6, and IL-1β release post- lipopolysaccharide (LPS) stimulation, increasing the risk of adverse events and death in sepsis [112,113,114].

Transcriptomic analysis using high-throughput multiplex mRNA sequencing has shown an increased expression of genes related to anti-inflammatory cytokines and inhibitory signaling molecules in sepsis, contrasting with the downregulation of pro-inflammatory cytokines and mediators such as T TLRs [115,116]. Notably, less than 5% of monocytes in septic patients can produce cytokines post-stimulation, a striking contrast to the 15–20% in healthy controls [117].

Single-cell RNA sequencing (scRNA-seq), particularly of leukocytes, is a powerful tool to delineate immune cell responses in sepsis. This approach, distinct from bulk RNA sequencing, offers insights into individual cell states. Recent studies have utilized scRNA-seq of peripheral blood mononuclear cells (PBMCs) from sepsis patients, identifying four distinct monocyte states (MS1–4) [116]. MS1 cells (CD14^−^) are marked by high levels of resistin, arachidonate 5-lipoxygenase activating protein, and interleukin-1 receptor type 2 (IL1R2). MS2 cells express high levels of class II major histocompatibility complex (MHC-II), MS3 cells are non-classical CD16^high^ monocytes, and MS4 cells exhibit low class II MHC and cytokine expression. Expanded MS1 cells, associated with sepsis, suggest a link between mitochondrial dysfunction and immune paralysis, as evidenced by their lower response to LPS compared to healthy controls [118].

#### 3.1.3. Natural Killer (NK) Cells

The particular tissue distribution and the low number of CD56^+^CD16^+/−^ NK cells in peripheral blood make studying this cell subset challenging, especially in the context of human sepsis, and can explain, at least in part, why such cells have not been deeply investigated [119,120]. However, they are abundant in some tissues like the lungs [121,122] which are particularly prone to dysfunctions in ICU patients.

The initial phase of sepsis is characterized by an increased number of activated NK cells expressing high levels of TLR-2/4/9 and CD69 [123], as well as high plasma concentration of granzyme A, interferon (IFN)-γ, and IL-12p40 [119,124,125]. Furthermore, in the hyperinflammatory stage, the enhancement of inflammatory factors leads to abnormal NK cell activation that can trigger a cytokine storm through a positive feedback loop resulting in severe organ damage [126,127]. However, prolonged hyperactivation leads to impaired NK cell function underlined by downregulation of cytotoxic genes [128] and overexpression of the inhibitory molecule NKG2A [129,130].

Nevertheless, in septic patients, the number of circulating NK cells was significantly decreased, especially in patients with infections due to gram-negative bacteria [131,132], and it was associated with increased mortality [133]. The mechanism at the basis of NK reduction in the bloodstream is not fully understood, but it is likely that an increased tendency to apoptosis might promote this phenomenon. However, animal studies revealed that there can be a rapid migration of NK cells to the site of infection within 4–6 h; thus, this might promote NK cell decline from blood circulation [134]. Therefore, NK dysfunction and defects may be a feature of the immunosuppressive state during the late phase of sepsis.

NK cells’ reduced potency of cytokine production (i.e., IFN-γ, IL-23) in polymicrobial sepsis suggest their tolerance to TLR agonists [120,135,136,137]. As NK cells play an integral role in antiviral defense, impaired NK cell function may lead to the reactivation of latent viruses, like CMV, that is commonly reported among ICU patients [119,138,139,140]. In COVID-19 patients, however, the absolute number of cytokine-producing CD56^bright^ NK cells or cytotoxic CD56^dim^ NK cells decreased with enhanced NK cell activation characterized by Ki-67, HLA-DR, and CD69 [141], which indicated sepsis [142].

The recent observations on NK cells suggest a potential application of NK cell therapy for the treatment of sepsis, but no current studies provide exhaustive mechanistic insights into their function during sepsis for different reasons. First, measuring different parameters in blood NK cells from patients with sepsis cannot provide a detailed picture of all tissues and compartments where NK cells alter their phenotypic or functional characteristics. Second, several clinical studies have small sample sizes. Third, few parameters have been measured in the early phases of sepsis. Fourth, the population of patients under analysis has high heterogeneity in terms of the severity of the infection and type and site of infection. All of these limitations make it difficult to determine the real role of NK cells in the pathogenesis of sepsis. Future studies have to address this challenge by taking into account the aforementioned limitations.

#### 3.1.4. γδ. T Cells and MAIT Cells

Sepsis affects γδ T cells, which recognize lipid antigens from various pathogens and primarily reside in the intestinal mucosa. These cells respond rapidly to a stimulus through IFN-γ and IL-17 release [143,144]. It has been shown that IL-17^+^ γδ T cells play an important role in polymicrobial sepsis [145], especially in lung failure [146]. Moreover, γδ T cells in septic patients showed lower levels of CD69 and IFN-γ expression after ex vivo stimulation, more so in non-survivors than survivors [147].

Significant depletion of γδ T cells in septic patients, due to increased apoptosis especially in CD3^−^CD56^+^ γδ T cells subset [148], correlates with higher illness severity and mortality [149,150]; also, their loss in the intestinal mucosa can lead to pathogen invasion and secondary infections.

Mucosal-associated invariant T (MAIT) cells are characterized by the expression of a semi-invariant αβ T-cell receptor (TCR) that differs from other conventional T cells. MAIT cells recognize and respond to various bacterial and fungal metabolites presented on the major histocompatibility complex class 1-related protein (MR1) [151,152,153] and play a main role in the response to different tumors [154,155]. The TCR expressed by MAIT cells is constituted of an invariant domain of the α chain consisting of TRAV1, corresponding to Vα7.2. Beside the expression of the aforementioned TCR, MAIT cells can also be identified by high expression of CD161 in humans [151,156]. Effector functions of MAIT cells, as well as their number in the bloodstream, are impaired during sepsis [152], which may reflect a re-distribution into the inflamed tissue, as shown for human peritonitis [157], and the deficiency of MAIT cells increases sepsis-related mortality in mice models. Also, downregulation of either *Ifng* mRNA or IFN-γ production occurs in septic mice [158]. In septic patients, an upregulation of activation markers like CD69, CD38, CD137 occurred, which return to normality during recovery [158]. Also, during sepsis, HLA-DR is upregulated in MAIT cells, and survivors present higher levels of HLA-DR and PD-L1 in comparison to non-survivors [159].

### 3.2. Adaptive Immunity

#### 3.2.1. T Lymphocytes

Sepsis has marked effects on tissue and circulating T lymphocytes, particularly CD4^+^ T cells. CD4^+^ T helper (Th) cells facilitate immunological processes in the body by assisting other cell types, such as B cells, during differentiation, activating cytotoxic T cells, and stimulating monocytes [97,160,161,162,163,164,165]. Based on the type of cytokines that mature CD4^+^ Th cells produce upon stimulation, mature Th cells have been categorized into Th1, Th2, and Th17 subsets. Upon stimulation, the aforementioned Th subsets elicit the production of characteristic effector cytokines: IFN-γ for Th1, IL-4 for Th2, and IL-17 for Th17 T cells. These cytokines confer protection against specific pathogens, such as intracellular viruses/bacteria, helminths, and fungi [166]. Each CD4^+^ T cell subset is typically characterized by the expression of subset-specific transcription factors (TFs), which facilitate lineage-specific gene expression and cytokine secretion. Unfortunately, purifying cells through flow cytometry methods based on the intracellular detection of TFs, which requires cell fixation and permeabilization, inhibits most downstream analyses. Therefore, researchers developed strategies using the combinatorial expression of surface chemokine receptors or other surface markers to identify various subsets of CD4^+^ T cells. The expression of T-bet and the chemokine receptor CXCR3 feature in the Th1 subset, while Th2 is characterized by the expression of GATA-3 and CCR4 chemokine receptors, and Th17 is identified by the expression of both CCR6 and FOXP3 [166].

One of the most damaging consequences of septicemia is the development of apoptosis that dramatically reduces CD4^+^ cell populations [167,168]. Patients who did not survive to sepsis had a significantly greater incidence of apoptosis in lymphocytes, particularly CD4^+^ cells, compared to those who survived [167]. Among CD4^+^ cells, the production of Th1- and Th2-related cytokines decreases both during and after sepsis [169]. Significant reductions in the transcription factors T-bet and GATA3, which regulate the Th1 and Th2 responses, respectively, provide evidence that the above-mentioned CD4^+^ subsets are suppressed in the course of sepsis [170]. The reduction of Th17 cytokine production in patients with sepsis was likely to have a harmful effect on long-term survival and may increase their vulnerability to fungal infections that are often seen in critically ill populations [171]. Additionally, the administration of IL-7 enhanced Th17 cell responsiveness, leading to a reduction in mortality associated with secondary fungal infections [172].

When T cells do not undergo apoptosis after sepsis, their phenotype and functions are greatly impaired [173]. CD4^+^ T cells in septic patients upregulate inhibitory markers like cell immunoglobulin and mucin domain-containing protein-1 (ICAM-1, or CD54), lymphocyte activation gene 3 (LAG3, or CD223), and cytotoxic T-lymphocyte-associated protein 4 (CTLA-4, or CD152) as well as exhaustion markers such as PD-1 (CD279) and its ligand PD-L1 (CD274) [10,174], decreased production of pro-inflammatory cytokines like IL-2 and IFN-γ, decreased STAT-5 phosphorylation upon TCR activation, decreased CD3 and CD28 expression, and alterations of Th subsets with a shift towards the Th2 profile of cytokine secretion [54,112,113,175]. Also, PD-1 upregulation seemed to be correlated with diminished T cell proliferation and increased nosocomial infections and mortality [102]. Indeed, in COVID-19 patients, T lymphocytes showed a reduced proliferative ability in vitro [176].

Sepsis also results in the dysfunction of CD4^+^ T cells due to metabolic reprogramming. T lymphocytes from patients with sepsis exhibit altered basal metabolic content due to reduced levels of ATP and inhibition of the mitochondrial oxidative phosphorylation system (OXPHOS) and glycolysis pathways [173]. 

Meanwhile, the CD8^+^ T cells are responsible for clearing an infection and producing memory CD8^+^ T cells as a result of infection or vaccination. Secretions of cytokines (including IFN-γ and TNF) and the ability to lyse cells are the main effector functions of CD8^+^ T lymphocytes [177]. Here, there are several dysfunctions associated with CD8^+^ T cells, including increased apoptosis, cell exhaustion, decreased proliferation, and cytotoxic activity [178]. Monocytes can induce CD8^+^ T cell anergy through the membrane expression of sialic acid-binding Ig-like lectin 5 (SIGLEC5, or CD170), which is an immune checkpoint that also exists in soluble form. SIGLEC5 plays a specific role against CD8^+^ but not CD4^+^ T cells. Thus, its expression on monocytes is higher in septic patients than in healthy volunteers after ex vivo LPS stimulation. In addition, SIGLEC5 restrained CD8^+^ T cell proliferation in vitro without inducing apoptosis [179].

There is also another phenotypic change in CD8^+^ T cells from septic patients. In a study, the number of naïve CD8^+^ T cells decreased and the number of effector/effector memory CD8^+^ T cells increased in septic patients in the first 24 h of ICU admission. Furthermore, the naïve CD8^+^ T cell compartment was also restored in the septic individuals when blood collection occurred 28 days after ICU admission. Sepsis, therefore, can significantly alter the naïve CD8^+^ T cell composition and directly affect the host’s ability to respond to new infections. These findings may have direct implications for therapeutic interventions to improve CD8^+^ T cell function in immunocompromised patients [180]. Immunocompromised individuals with sepsis are more likely to experience reactivation of latent viruses [181,182]. A study examining the activity of T cells in sepsis patients with human cytomegalovirus (HCMV) found that CD8^+^ T cells had impaired polyfunctionality [183] (p. 201). Additionally, patients who experienced HCMV reactivation had a significantly reduced frequency of CD8^+^ T cells [183]. Patients with viral reactivation showed increased PD-1 expression on their T cells compared to those without [183]. This is another example of T cell exhaustion. Polyfunctional CD8^+^ T cells were inversely proportional to the expression of PD-1 [183].

In adaptive immunity, T regulatory cells (Tregs) play a major role in maintaining self-tolerance and suppressing autoimmune disease by inhibiting the responses of other effector T-cell subsets. Treg cells exhibit greater resistance to sepsis-induced apoptosis when compared to other T cell subpopulations [184]. In cases of sepsis, critical illness, and inflammation, Tregs enhance the harmful suppression of effector T cells, resulting in extended recovery periods and heightened risk of complications [185]. T lymphocyte dysfunction after sepsis is characterized by an increased proportion of a CD4^+^CD25^+^CD127^low^ regulatory T cell subpopulation among circulating patients’ lymphocytes due to a decrease of effector T cell numbers [186].

The potential implication of an augmented quantity of circulating Tregs, specifically CD4^+^CD25^+^Foxp3^+^ cells, may be involved in the promotion of lymphocyte anergy and immunoparalysis linked to sepsis [58]. In fact, although pro-inflammatory effector Th cells have lost their cytokine production capacity, the capacity of Tregs of producing IL-10 can remain constant [187]. Furthermore, in critically ill patients, an increased proportion of Treg is linked to a higher risk of subsequent nosocomial infections. Additionally, it is possible to include this parameter in a panel that can be useful for classifying patients in the context of a precision medicine strategy [188].

#### 3.2.2. B Lymphocytes

There is limited information about the B lymphocyte response in septic patients. As observed for Tregs, there can be a decrease in the absolute number of circulating B cells [131] but an increase in their relative percentage among total lymphocytes [35,124,131,133,168,189,190,191,192,193,194,195,196] (p. 2). Even if the reduction in the number of B cells and their subsets in sepsis remains debatable, B cell count in non-survivors was significantly lower compared to survivors [35,189,195,196,197,198,199,200].

The mechanism at the basis of B cell depletion is not fully understood, although accelerated apoptosis has been hypothesized, since previous evidence has revealed that B cell apoptosis occurs in blood, intestine, and peripheral lymphoid organs in patients with sepsis and animal models [201,202,203,204].

B cell phenotypes change in sepsis, with a marked reduction of CD19^+^CD23^+^ activated regulatory B cells and CD19^+^CD5^+^ natural responder B1a cells, although the number of CD19^+^CD69^+^ early activated B cells remains unaffected [191]. B1a cells are able to secrete IgM [205,206] as well as immunomodulatory cytokines like IL-10, spontaneously or after infection [206,207,208], and GM-CSF, IL-6, IL-3, TNF [206,207]; therefore, there is evidence that B1a cells play a protective role in sepsis and survival benefits [209].

Whilst the number of total B cells decreases, the proportion of B cell among total lymphocytes increases and the proportion of circulating CD21^low^CD95^hi^ exhausted-like B cells augments [189,192] along with the upregulation of PD-1, PD-L1, CD95, and CD80, that not only impair the antigen-presenting activity of B cells but also affect prognosis [191,193,200] (p. 2). Unlike T cells, there is a decrease of IL-10-producing Breg cells [210] which is correlated to poor prognosis [211]. Also, the reduction of the percentage of CD19^+^CD24^hi^CD38^hi^ Breg cells is associated with development of septic shock [211].

As well as their phenotype, B cell function is also impaired in sepsis, with a reduced ability of proliferation and decreased ability of antibody secretion because of insufficient IgM and IgG synthesis [192,195,212]; indeed, hypogammaglobulinemia is common in sepsis [213,214] (p. 202). In particular, a significant decrease of IgM levels is associated with worse outcome [197,198] since lower levels of IgM have been found in non-survivors compared to survivors [212]. However, IgM deficiency is probably not due to the impairment of Ig class switching, but instead occurs because of the depletion of CD27^+^CD21^hi^ resting memory B cells [35,195]. In elderly people with sepsis, a reduction of immunocompetent B cells with the augment of CD21^−/low^ exhausted B cells is developed along with insufficient IgM production, which may increase susceptibility to secondary infections [192]. In fact, memory B cells are more susceptible to sepsis-induced cell death [215] triggered by the extrinsic pathway induced by receptors like Fas (CD95) and by the intrinsic pathway mediated by mitochondria [202,204,216,217].

Furthermore, in sepsis, B cells can produce a variety of pro-inflammatory cytokines like IL-6, TNF, IL-3, and GM-CSF to reinforce the systemic inflammatory response [218,219,220]. These cytokines also play an important role in polarizing CD4^+^ T cells towards the Th1 and Th17 phenotypes and in activating innate immune cells like monocytes [221,222].

### 3.3. Metabolic Shifts in Immune Cells

For both their housekeeping functions and specialized activities, immune cells require substantial amounts of energy. Glucose is metabolized via glycolysis or oxidative phosphorylation (OXPHOS) to produce ATP. OXPHOS is an oxygen-dependent process that takes place in the mitochondria and is highly efficient, producing a total of 36 molecules of ATP from each molecule of glucose. Glycolysis, on the other hand, is oxygen-independent and takes place in the cytosol, where one molecule of glucose is broken down into two molecules of pyruvate; this pathway is less efficient but can be mobilized rapidly [223,224]. Sepsis is initially characterized by a hypermetabolic state, featuring increased ATP production, respiration, and release of different hormones. Subsequently, a hypometabolic state ensues, characterized by reduced OXPHOS, ATP production, and endocrine hormone release or resistance [225,226]. This transition forms the basis of the “hibernation theory”, which posits a protective mechanism allowing cellular recovery by reducing metabolic demands in an environment of impaired ATP production [227]. A shift towards glycolysis as the primary energy-producing pathway occurs, with lactate—a by-product of anaerobic metabolism—serving as a clinical marker for sepsis diagnosis, despite its controversial predictive value [228,229,230].

Sepsis-induced mitochondrial damage is the main cause of metabolism disorders in septic patients [231]. Even though the mechanism at the basis of mitochondrial dysfunction is not completely understood, several hypotheses were made, including a possible involvement of pro-inflammatory cytokines, ROS, and NO, whose production is upregulated in sepsis [232,233]. Indeed, ROS and NO might interfere with the mitochondrial electron transport chain (ETC), hence affecting cellular respiration [227,234,235]. Moreover, a reduced expression of genes encoding for proteins involved in ECT [232,235,236] as well as lower copies of mitochondrial DNA (mtDNA) and a decrease of metabolic rate can play a role [237]. Also, hormonal changes may contribute to the reduced ability to produce ATP through OXPHOS [238]. While ETC dysfunction reduced its efficacy, it enhances the possibility of ROS generation, so it has been hypothesized that a shift towards glycolysis is set up as a protective adaptive mechanism [239]. Thus, a 67% increase in glucose uptake and an almost twofold increase in GLUT1 expression have been observed during sepsis in animal models [240] (p. 199).

Glycolysis in immune cells leads to the production of pro-inflammatory factors, causing “inflammatory storms” and subsequent cellular damage. Patients may survive these storms but then deteriorate due to ensuing immunosuppression. The development of immunosuppressive conditions often involves reduced expression of genes for pro-inflammatory cytokines (e.g., TNF, IL-6, CCL2) and increased expression of anti-inflammatory cytokines (e.g., IL-4, IL-10), impeding effective combat against secondary infections [241,242].

This immunometabolic shift from glucose to fatty acid oxidation is marked by increased levels of fatty acid transporters (like CD36) and CPT-1 in leukocytes [243]. SIRT1 and SIRT6 facilitate this metabolic transition and link metabolism with the acute inflammation response phases. Fatty acid oxidation (FAO) not only promotes an anti-inflammatory phenotype but also produces lactate during the pro-inflammatory phase, exerting a potent immunosuppressive effect [243].

## 4. Sepsis Therapy: Phenotyping and Personalized Approaches

Sepsis presents significant challenges in diagnosis and treatment due to its heterogeneity that often leads to difficulties in designing successful clinical trials and variable diagnoses among critical care experts, particularly in cases where symptoms are masked by comorbidities, or when laboratory evidence is unclear [11,19,244,245,246]. The concept of precision medicine is gaining traction in response to these challenges, focusing on tailored strategies based on individual patient characteristics. This approach is crucial given the varied manifestations of sepsis and the need to determine effective treatment strategies for different patient subgroups at the optimal times in the disease process [247,248]. In the ICU, the diversity in patients’ clinical histories and conditions, and their responses to treatments, especially in sepsis, highlights the need for a more nuanced understanding of individual characteristics. Recent guidelines aimed at standardizing care for sepsis and septic shock patients have been discussed for potentially oversimplifying the complexity of sepsis management, suggesting that a singular strategy may not be suitable for all patients [249,250,251,252]. This complex scenario underscores the emerging role of cell phenotyping in sepsis, which involves categorizing patients based on observable characteristics, such as clinical presentation and first outcomes. Understanding these phenotypes could aid in tailoring treatments more effectively and could potentially lead to the discovery of novel therapeutic targets and specific biomarkers that might respond differently to various treatments [253,254,255].

However, identifying these phenotypes is challenging, primarily due to the lack of empirical evidence on how to individualize treatment based solely on clinical variables. Artificial intelligence (AI) and machine learning have been making significant inroads in this area. These technologies analyze large data sets to identify patterns and patient groups, proving invaluable in recognizing different disease expressions in sepsis. Clustering analysis, a form of unsupervised machine learning, has been used to identify novel patient subgroups or phenotypes based on multiple variables [256,257]. Nevertheless, these methods have limitations, such as biases in training datasets and indeterminacy in the optimal number of clusters. Therefore, findings from machine learning need rigorous evaluation and external validation for real-world applicability [258,259,260,261,262]. Temperature phenotyping in sepsis, for instance, has shown that a significant portion of septic patients remain hypothermic, linked to increased mortality, whereas fever is associated with decreased mortality. This variation in body temperature was investigated in large cohorts, revealing additional phenotypic characteristics and associations [244,263]. Hemodynamic traits and multiorgan dysfunction have also been used for phenotyping septic patients, identifying distinct phenotypes based on factors like systolic blood pressure and organ dysfunction patterns. These phenotypes could guide more personalized treatment approaches in sepsis, such as tailored fluid responsiveness and vasodilator therapies [264,265,266,267,268,269,270]. Furthermore, the impact of fluid management on sepsis outcomes has been noted, with different phenotypes responding variably to fluid resuscitation. Identifying phenotypes prone to fluid overload could optimize fluid management strategies, potentially impacting patient outcomes [271,272,273]. Ongoing research in sepsis phenotyping is considering various factors like the site of infection, organ dysfunction, and specific diseases like COVID-19 and ARDS, aiming to correlate phenotypes with treatment responses and outcomes [274].

In conclusion, the path towards effectively managing sepsis lies in the implementation of precision medicine, where immune therapy is tailored to individual patient characteristics and pathophysiological changes. Despite centuries of understanding and decades of clinical definitions, sepsis remains a complex and ill-defined syndrome. The future of sepsis treatment will likely involve a nuanced approach, depicted in Figure 3 (adapted from Sinha et al. [115]), based upon a combination of data that considers both clinical phenotypes and immunological profiles, to guide therapy in a manner that is both time-sensitive and individualized [275,276,277].

In the evolving landscape of sepsis therapy, the move towards personalized medicine reflects a deeper understanding of the condition’s complexity. Sepsis indeed underscores the inadequacy of one-size-fits-all treatment approaches. The push for individualized therapies is rooted in the recognition of sepsis’s diverse manifestations and the necessity to tailor treatments for different patient subgroups at the most effective stages of the disease process [248,249,250,251,252,253,254,255,256,257,258,259,260,261,262,263,264,265,266,267,268,269,270]. The heterogeneity among sepsis patients results in divergent outcomes despite similar interventions. This variation challenges the universal applicability of standardized care protocols like those proposed by the Surviving Sepsis Campaign by the Society of Critical Care Medicine (SCCM) and the European Society of Intensive Care Medicine (ESICM) [249,250,251,252]. Consequently, there is an increasing call for a nuanced approach that considers each patient’s unique characteristics in sepsis management.

A crucial aspect of personalized sepsis therapy involves the dynamic balance between mitigating excessive inflammation and boosting the host’s immune response. This balance is pivotal and varies depending on several factors, including age, overall health status, and the stage of sepsis. For instance, older patients, who are more susceptible to developing immunosuppression (or to an increase in immunosuppression due to their age), may benefit from immune-boosting strategies, while those in the initial hyperinflammatory phase might need anti-inflammatory interventions. The timing of these treatments is critical, as patients often transition from a state of hyperinflammation to immunosuppression during the course of sepsis [163]. Recent advances in sepsis research have facilitated the identification of gene expression-based subphenotypes, aiding in the development of clinical classifiers and underscoring the molecular heterogeneity of the disease. Studies by Wong et al. [245,278], Scicluna et al. [116], and Sweeney et al. [279] have revealed various sepsis phenotypes such as “inflammopathic”, “adaptive”, and “coagulopathic”, paving the way for personalized treatment approaches [115].

Flow cytometry has become a valuable tool in sepsis research, particularly for quantifying HLA-DR on monocytes, a key marker of immunosuppression in critically ill patients. This technique not only correlates with the ability to mount an effective immune response but also aids in distinguishing sepsis from other conditions like SIRS and in understanding the chronic and treatment-resistant aspects of sepsis [30,280]. The journey towards effective molecular targeted therapies in sepsis has been challenging. Initial optimism based on preclinical studies targeting TNF was dampened by clinical trials that revealed increased mortality with anti-TNF antibodies [19,281]. Similarly, the role of corticosteroids in sepsis treatment remains contentious, with early reports of reduced mortality in septic shock not being consistently replicated in subsequent trials [282,283].

One of the major challenges in developing effective therapies is the inherent heterogeneity of sepsis patients. Researchers have noted that the broad clinical criteria used to define sepsis result in a patient group so diverse that finding universally effective therapies becomes a daunting task [284]. 

This heterogeneity stems from variations in patient factors such as age, sex, comorbidities, lifestyle, genetics, site of infection, and the specific pathogen involved [285]. To address this, strategies to reduce heterogeneity in clinical trials have been suggested, such as limiting enrollment based on specific criteria or utilizing biomarkers to target the underlying pathophysiology. While this approach increases the precision of patient selection, it also raises research costs and limits the generalizability of results [286]. The complexity of molecular pathways in sepsis further complicates therapeutic development. Even modest stimuli can elicit extensive and dynamic changes in genes and metabolites, suggesting that targeting a single pathway in sepsis may be insufficient for significant clinical improvement [287,288]. The use of preclinical models, typically involving young, healthy animals with minimal supportive care, also poses challenges for translating findings to human sepsis cases. Differences between species, such as mice’s resistance to endotoxin, contribute to the discrepancies between preclinical successes and clinical trial outcomes [289,290].

Emerging treatments for sepsis are increasingly focusing on optimizing fluid, hemodynamic, and sedative management. There is a shift towards boosting immunity during later stages of immunoparalysis and enhancing organ function through approaches such as cell-based therapies [291,292,293]. The development of new guidelines and definitions, such as the Sepsis-3 definition by the SCCM and the ESICM, reflects the evolving understanding of sepsis and aims to improve its identification and management [294].

In conclusion, the pathogenesis and treatment of sepsis demand a multifaceted approach. Despite the failure of many targeted therapies in clinical trials, advancements in supportive care have led to improved outcomes. Future directions in sepsis research should focus on novel therapeutic pathways targeting organ dysfunction and refining clinical and biological criteria for patient selection in clinical trials. The shift towards personalized medicine in sepsis treatment is crucial, considering the complexity and heterogeneity of the disease. This approach, combining clinical phenotypes with immunological profiles, holds promise for guiding more effective and individualized therapies (Figure 4).

## 5. Omics Technologies

The evolving field of “omics” (genomics, transcriptomics, proteomics, metabolomics, cytomics) offers systems-biology-based approaches for developing new diagnostic tools in sepsis [295]. With increased computing power, deep phenotyping in critical care settings has become more feasible, aiming to identify homogeneous subgroups within otherwise heterogeneous populations using multidimensional data [115].

Phenotyping can categorize patients into subgroups of a clinical syndrome or disease, defined by a shared feature. Endotyping goes further, categorizing patients into subtypes associated with specific pathobiological or functional mechanisms [57]. Bulk transcriptomic approaches have been used to investigate phenotypes in adult sepsis patients, revealing two sepsis response signatures (SRS1 and SRS2) using unsupervised clustering algorithms. SRS1, associated with poorer outcomes, is enriched with transcripts related to cell death and immune exhaustion [296]. However, transcriptional phenotyping is challenged by variability in detected gene clusters across different methods and populations.

Epigenetics may also play a role in understanding the pathogenesis of sepsis-induced immune dysfunctions, particularly regarding DNA methylation and histone acetylation of inflammatory genes [297,298]. Micro-RNAs (miRNAs) and small non-coding RNAs (sncRNAs) modulate gene expression, with specific miRNAs, like miR-15a, miR-16, miR-122, miR-133, miR-193, miR-223, and miR-483–5p being potential biomarkers in sepsis [299,300,301].

Growing attention has recently been given to circulating extracellular vesicles (EVs), which are involved in cell-to-cell crosstalk through delivery of proteins, lipids, and genetic materials. EVs can be investigated in plasma through flow cytometry [302] and are also potential mediators of sepsis-induced endothelial injury and myocardial dysfunction [303,304]. The number of EVs increases during sepsis [305,306] and it is positively correlated with the severity of the disease [305]. In particular, in the early stage of sepsis, acute phase reactive proteins and Ig involved in inflammatory response are upregulated [307]. 

Also, levels of mRNA involved in antioxidant defense and oxidative stress are upregulated in the EVs of septic patients [308], that also show an upregulation of DNA methyltransferase (DNMT) 1, DNMT3A, and DNMT3B mRNA, which is correlated with poor prognosis [305]. Neutrophil-derived EVs mainly have anti-inflammatory effects [309,310,311,312], with upregulation of alpha-2-microglobulin and ceruloplasmin, which promote neutrophil adhesion, enhance bacterial clearance, reduce inflammatory response, and improve survival in septic mice [313,314]. On the other hand, host-derived EVs carry a variety of PAMPS and DAMPs like high mobility group box 1 (HMGB1), extracellular cold-inducible RNA-binding protein (eCIRP), mtDAMPs that promote inflammatory responses, neutrophil migration, and organ damage [315,316,317,318,319,320,321,322,323] (p. 20).

Thus, in sepsis, EVs play a range of regulatory roles in either the pathogenesis or treatment of sepsis, so they might serve as innovative biomarkers, together with all the alterations found in septic patients (summarized in Table 1), for predicting and monitoring sepsis progression. As omics technology has developed and been applied, more studies have shown that cargo carried by EVs in sepsis is expressed dynamically, and this correlates with specific clinical features of protein expression, RNA expression, and metabolic changes in EVs.

## 6. Future Directions

While omics technologies herald a new era of precision medicine in sepsis care, offering nuanced patient stratification, the immediate need for practical diagnostic and therapeutic tools in clinical settings is pressing. In this context, as illustrated in Figure 5 (which shows the phenotype and quantification of MAIT cells in septic patients and healthy donors), flow cytometry emerges as a crucial tool, enabling rapid and precise immunophenotyping. This technique’s ability to delineate specific immune cell populations and their activation states provides valuable insights into the immune dysfunctions characteristic of sepsis [30,324]. The utility of flow cytometry is contingent upon the standardization of protocols, ensuring consistent and reproducible results across different laboratories and clinical studies. This standardization is vital for the multi-center evaluation of immunological biomarkers and comparable patient stratification, which are essential for advancing personalized medicine in sepsis care [325].

The integration of advanced machine learning techniques with multi-omics datasets, including genomics, proteomics, and metabolomics, also offers substantial progress in sepsis research. Support Vector Machines (SVMs) and ensemble methods such as Random Forests have proven effectiveness in identifying novel biomarkers and immune profiles from the complex, high-dimensional omics data, which are critical for early diagnosis and the development of personalized therapeutic strategies [326]. These algorithms excel in classifying patients based on subtle, multidimensional patterns detected within omics data, that may not be detectable using traditional statistical methods.

In the field of Artificial Intelligence, deep learning methods like Convolutional Neural Networks and Recurrent Neural Networks have gained traction for their ability to uncover and interpret intricate spatial and temporal patterns within omics datasets. A prediction tool named PENCIL, which utilizes the Learning with Rejection (LWR) strategy, has been developed to select informative features and identify cell subpopulations from single-cell data efficiently. PENCIL is capable of analyzing one million cells in one hour and has been used to identify cancer patients responding to immune checkpoint blockade therapy on the basis of immune cell subpopulations and clinical variables [327]. For instance, this method could be applied to predict sepsis outcome.

Another recent innovation is Stabl, a machine learning approach that provides a unified framework for supervised learning to identify a sparse, reliable set of biomarkers across various single-omics and multi-omics datasets. Starting from 1400–35,000 features in a dataset, it extrapolates 4–34 putative biomarkers. Hence, Stabl has been applied to integrates multi-omics data of pre-term birth and a pre-operative immune signature of post-surgical infections and to predict labor onset [328]. Thus, it might be exploited in sepsis settings to predict sepsis outcomes, as well as biomarkers associated with dysfunctional immune systems. Finally, NAVOY is a real-time sepsis prediction algorithm that uses 4 h of routinely collected laboratory values and clinical parameters as input to identify patients with risk of developing sepsis three hours before sepsis onset [329].

By analyzing variations in patient-specific immune responses, these models can suggest personalized and combined treatment strategies that are optimized for efficacy while minimizing adverse effects. This tailored approach to sepsis treatment represents a significant move towards precision medicine, harnessing the full potential of AI and omics data in clinical settings.

In conclusion, the integration of clinical data, advanced laboratory results, and the use of machine learning algorithms is now paving the way to personalized and targeted therapy.

## 7. Conclusions

Clinical phenotypes of septic patients are heterogeneous due to various factors such as age, gender, comorbidities, site of infection, and the specific pathogen involved, and are difficult to identify and classify. Thus, even if the treatment of sepsis, based on standardized clinical procedures characterized by the administration of broad-spectrum antibiotics, fluid therapy, and vasopressors, significantly reduces mortality, an improvement in its efficacy is still required.

In the era of multi-omics techniques, the identification of key changes in septic patients is critical to help clinicians determine the most effective diagnostic and therapeutic approaches for each individual, moving beyond one-size-fits-all solutions. The fine analysis of immune system alterations is now feasible by using different single-cell methods such as flow cytometry, which can simultaneously detect up to 50 proteins and scRNA-seq. Furthermore, the study of metabolic changes and of the cell–cell crosstalk is helping to provide a clearer scenario of sepsis phenotypes and its immune phase.

In last years, AI and machine learning made significant inroads in this area, enabling the analysis of large datasets to identify patterns and patients’ groups, through algorithms like Stabl and NAVOY and prediction frameworks like PENCIL.

The synergy of omics data and machine learning opens new perspectives in sepsis, not only supporting a better characterization of patients’ phenotypes, but also promoting a tailored approach to its treatment and towards personalized and targeted therapy.

## Figures and Tables

**Figure 1 cells-13-00439-f001:**
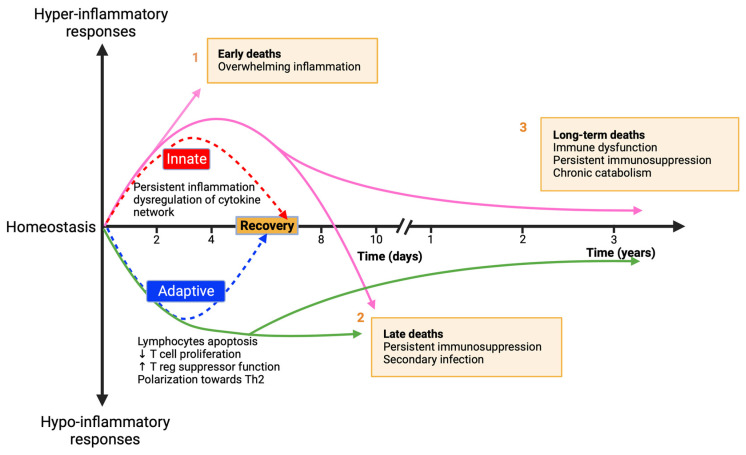
Sepsis Immune Response Timeline. 1: Early phase: intense immune activation causing high initial mortality due to “cytokine storm”. 2: Late phase: prolonged immunosuppression leading to secondary infections and organ failure. 3: Advances in ICU care have improved outcomes, yet late-phase mortality remains a challenge.

**Figure 2 cells-13-00439-f002:**
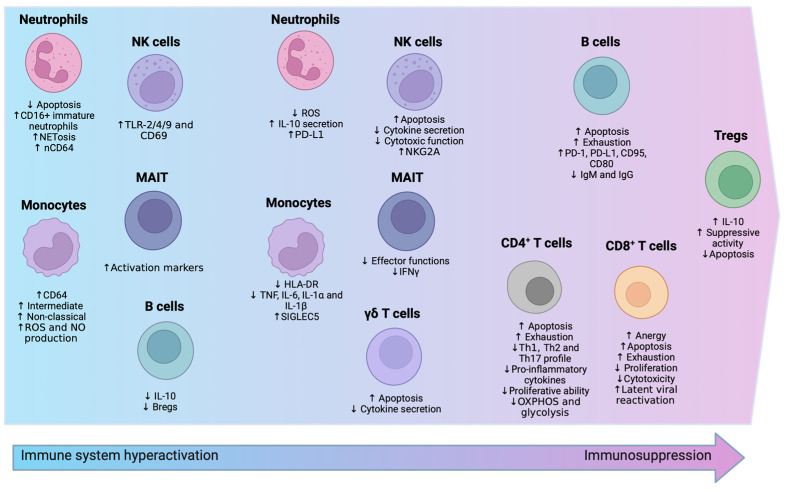
Immune Cell Modulations in Sepsis. The image provides a comparative overview of immune responses by different cell types during sepsis. The initial hyperactivation with reduced apoptosis in neutrophils and elevated activation in NK cells is depicted, along with increased apoptosis in γδ T cells and MAIT cells. Subsequently, it shows the transition to immunosuppression, with Tregs enhancing suppressive functions and B cells becoming exhausted. Both CD4^+^ and CD8^+^ T cells are characterized by increased anergy and apoptosis, signaling a state of immune exhaustion, which collectively portrays the biphasic immune landscape in sepsis.

**Figure 3 cells-13-00439-f003:**
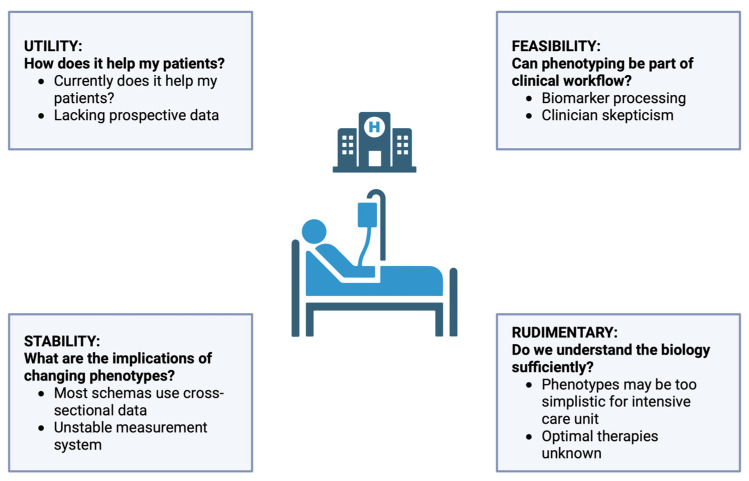
Evaluating Sepsis Phenotyping. This infographic details key considerations in sepsis phenotyping, divided into three aspects: UTILITY, questioning the immediate benefits for patient care; FEASIBILITY, addressing the practicality of incorporating phenotyping into clinical routines; and STABILITY, discussing the consistency of phenotypic classifications. A final note on RUDIMENTARY aspects suggests a need for deeper biological understanding to support phenotype-based treatments in intensive care settings.

**Figure 4 cells-13-00439-f004:**
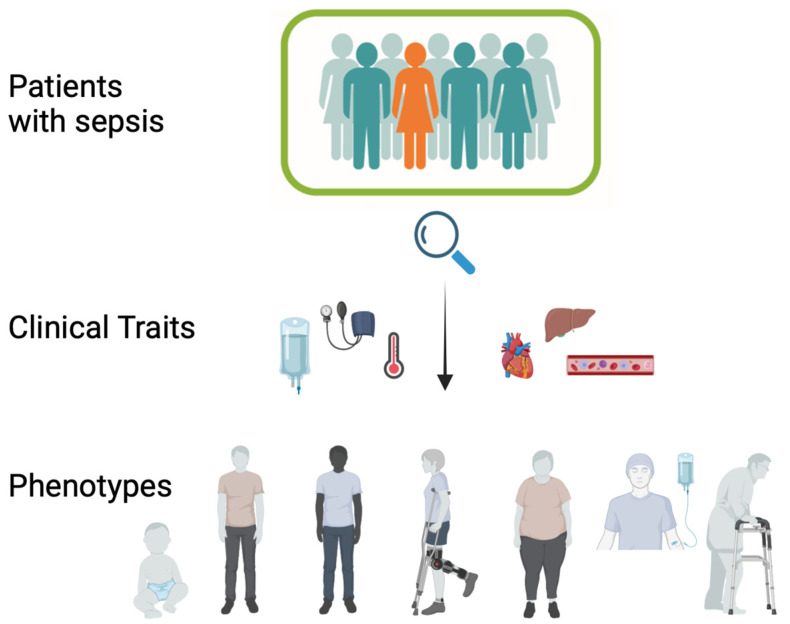
Clinical Phenotyping in Sepsis. The figure illustrates the categorization of sepsis into distinct phenotypes based on clinical traits like temperature, hemodynamics, and multi-organ dysfunction. It underscores the shift towards personalized medicine, highlighting the potential of combining clinical and immunological data to inform targeted therapies and improve outcomes in sepsis care.

**Figure 5 cells-13-00439-f005:**
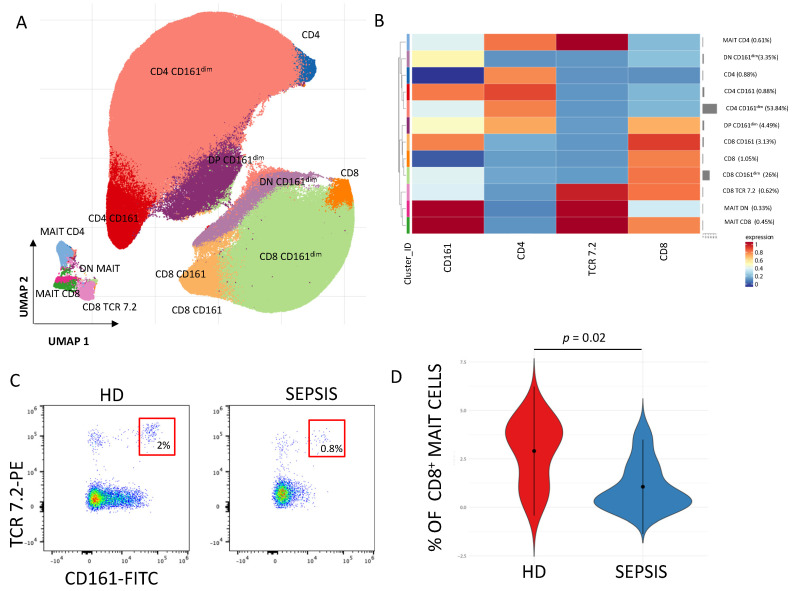
Quantification and phenotypic characterization of MAIT cells in septic patients. (**A**) The UMAP plot illustrates the 2D spatial distribution of CD3^+^ T cells across 15 samples, integrated with FlowSOM-based clusters. The color of the different clusters is reported on the left side of the heatmap. (**B**) Heatmap of the median marker intensities of the 4 lineage markers across the 12 cell populations obtained with FlowSOM algorithm (meta-20). The colors of cluster_id column correspond to the colors used to label the UMAP plot clusters. The color in the heatmap is referred to the median of the arcsinh marker expression (0–1 scaled) calculated over cells from all the samples. Blue represents lower expression, while red represents higher expression. The light gray line along the rows (clusters) indicates the relative sizes of the clusters. DP = double positive; DN = double negative. (**C**) Representative dot plot showing the percentages of CD8^+^ MAIT cells (within CD3^+^ T cells) in healthy donors (HD) and septic patients (SEPSIS). (**D**) Violin plot representing the percentage of CD8^+^MAIT cells in healthy donors (HD) and septic patients (SEPSIS) at time of admission in Intensive Care. Non-parametric student *t*-test; exact *p*-value is shown in the figure. [data produced by the Laboratory of Immunology, directed by Prof. Andrea Cossarizza, University of Modena and Reggio Emilia].

**Table 1 cells-13-00439-t001:** Main immune cells alterations, metabolic dysfunctions, and communication/cell-cell crosstalk that occur in septic patients.

Alterations in Sepsis	Main Findings in Septic Patients	Refs.
**Immune Cells Alterations**		
**Neutrophils**	reduced apoptosis and ROS production	[59,61,62,63,69,70]
	upregulation of CD64 and PD-L1 expression	[58,83,84,85,87,88,90]
	upregulation of CD16^+^ immature neutrophils	[63,64,65]
	enhanced IL-10 production and NETosis	[32,73,76,77]
**Monocytes**	enhanced CD64 expression	[104,105]
	reduced expression of HLA-DR	[94,95,96,97,98,99,100]
	upregulation of intermediate and non-classical phenotypes	[92]
	enhanced ROS and NO production	[104,108,109,110,111]
	upregulation of SIGLEC5 expression	[180]
	downregulation of TNF, IL-6, IL-1β, and IL-1α production	[106,107,112,113,114]
**NK cells**	enhanced apoptosis	[134]
	upregulation of TLR-2/4/9 and CD69 in the initial phase of sepsis	[123]
	upregulation of NKG2A expression	[129,130]
	reduced cytokine secretion	[120,135,136,137]
	impaired cytotoxic function	[128]
**γδ** **T cells**	enhanced apoptosis	[148]
	reduced IFNγ secretion and CD69 expression	[147]
**MAIT cells**	reduced effector functions	[152]
	upregulation of HLA-DR and PD-L1 expression	[159]
	reduced IFNγ production	[158]
	upregulation of CD69, CD38 and CD137 expression	[158]
**CD4^+^ T cells**	enhanced apoptosis	[167,168]
	enhanced exhaustion	[10,174]
	reduced Th1, Th2, and Th17 populations	[169,170,171]
	reduced production of pro-inflammatory cytokines	[112,113,174,175]
	reduced proliferative ability	[176]
	reduced OXPHOS and glycolysis	[172]
**CD8^+^ T cells**	augmented anergy	[179]
	enhanced apoptosis	[178]
	enhanced exhaustion	[178]
	decreased naive CD8^+^ T cells and increased effector/effector memory CD8^+^ T cells in the first phase of sepsis	[180]
	reduced cell proliferation	[178]
	reduced cytotoxicity	[178]
	increased reactivation of latent viruses	[181,182,183]
**Tregs**	augmented number of Tregs	[58,186]
	constant IL-10 production	[187]
	reduced apoptosis	[184]
**B cells**	enhanced apoptosis	[201,202,203,204]
	enhanced exhaustion	[189,192]
	upregulation of PD-1, PD-L1, CD95, and CD80 expression	[191,193,200] (p. 2)
	reduced IL-10 production	[210]
	reduced IgM and IgG production	[35,192,195,212]
	reduced number of Bregs	[211]
**Metabolic Alterations**		
**Mitochondrial dysfunction**	reduced expression of genes encoding for proteins involved in electron transport chain (ECT)	[232,235,236]
	lower copied of mtDNA	[237]
	decreased metabolic rate	[237]
	upregulated ROS production	[232,233,239]
**Glicolysis enhancement**	augmented lactate production	[228,229,230]
	increased GLUT1 expression	[240] (p. 199)
**Shift towards fatty acid oxidation (FAO)**	increased levels of CD36 and CPT-1	[243]
	SIRT1 and SIRT6 cordinate the metabolic swith toward FAO	[243]
**Communication/Cell-Cell Crosstalk**		
**miRNAs**	upregulated levels of miR-15a, miR-16, miR-122, miR-133, miR-193, miR-223, and miR-483-5p	[299,300,301]
**Extracellular vescicles (Evs)**	increased number of EVs	[305,306]
	EVs carrrying mRNA involved in antioxidand defence and oxidative stress	[308]
	upregulation of EVs carryng DNA methyltransferase (DNMT)1, DNMT3A, and DNMT3B mRNA	[305]
	neutrophil-derived EVs have anti-inflammatory effects	[309,310,311,312]
	pro-inflammatory response is enhanced by EVs carryng PAMPs, DAMPs like HMGB1, eCIRP, mtDAMPS	[315,316,317,318,319,320,321,322,323] (p. 20)

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
