# Peer review of "Advances and Challenges in Sepsis Management: Modern Tools and Future Directions"

_cells, 2024, doi:10.3390/cells13050439_

Round 1

Reviewer 1 Report

Comments and Suggestions for Authors

Dear Sirs,

this is a very interesting manuscript dealing with sepsis and its biomarkers. The role of immune cells in this context as well as the multi-omics approach are very welcome. The authors have very well presented their manuscript. My only comment is that although the references are too many, ie 331, only a very few have been in the last 5 years. Therefore, I suggest accept the manuscript in its current form, but with adding newer references in the referencing list.

Author Response

We thank the Reviewer 1 for comment. Even if we cited more than 100 papers, reviews and clinical trials published in the last 5 years, we changed and added some of the 331 references with more updated publications. All modifications are highlighted in green.

Reviewer 2 Report

Comments and Suggestions for Authors In this review, Santacroce et al. emphasized the potential clinical application of omics technologies and flow cytometry in patient-specific diagnosis and clinical management of septic patients. The review focuses on flow cytometry applications in immunophenotyping septic patients as a tool for the refinement of sepsis management. The manuscript is well-written and well organized.  However,  1. The introduction should be revised. As far as I am concerned, Singer et al. in “The Third International Consensus Definitions for Sepsis and Septic Shock (Sepsis-3)” (doi: 10.1001/jama.2016.0287) highlighted the limitations of previous definitions that included an excessive focus on inflammation, the misleading model that sepsis follows a continuum through severe sepsis to shock, and inadequate specificity and sensitivity of the systemic inflammatory response syndrome (SIRS) criteria. Also, the term “severe sepsis” was considered redundant. Authors should use the appropriate citation when referring to sepsis definition (lines 39-41). Since the authors mentioned the updated definitions for sepsis and septic shock, the entire introduction section should be revised accordingly. Moreover, authors should rephrase lines 265-266. Sepsis is an acute condition and it should not be characterised as chronic disease. 2. The authors could add a table including in the aggregate the main findings in septic patients. 3. The authors did not include a citation for the data presented in Figure 5. Comments on the Quality of English Language

Minor editing of English language required.

Author Response

We thank the Reviewer 2 for the comments.

  1. We amended as requested the introduction in accordance with the observations concerning the updated definition of sepsis (Singer et al., JAMA 2016). The term 'severe sepsis' has been rephrased. We corrected the citation in lines 39-41 and reworded the statement to refer to sepsis not as a chronic condition in lines 265-266.
  2. As suggested, we added a table (Table 1) that sums up the main alterations in septic patients, including the main findings in sepsis pathophysiology.
  3. Figure 5 represents unpublished data produced in our laboratory; therefore, we added “unpublished data” in the figure legend.

Reviewer 3 Report

Comments and Suggestions for Authors

This is a review involving changes in the innate immune system of septic patients.  It has been known for some time that the innate immune system in sepsis is disrupted.  As is well established, in such patients use of flowcytometry provides important information related to an inflated or depressed immune system and changes in the various subtypes of lymphocytes.  It has been known for many years that flowcytometry provides useful information using blood samples.  The review emphasizes the utility of flowcytometry in the setting of sepsis.  In general, this review contains very little new information.  The material presented is useful, but few new ideas are provided.

Author Response

We thank the Reviewer 3 for the comment. The findings presented in our review contribute to summarizing some of the main biomarkers that can be assessed through flow cytometry to understand immune dysfunctions in sepsis. Our report highlights the importance of phenotyping immune cells in the different phases of sepsis which might be considered an approach that has profound implications in the patient-specific diagnosis and therapeutic strategy. Indeed, our manuscript dives deep into the complexity of sepsis management, emphasizing the emerging role of flow cytometry in understanding immune dysfunctions and advancing personalized medicine. We believe that the review provides new insights into the application of flow cytometry in sepsis and highlights its potential in clinical settings, which is still too poor in terms of number of biomarkers analyzed at the same time.

However, in light of your comment, we emphasized an important technological advancement characterized by the integration of flow cytometry with omics data and machine learning (using algorithms like “PENCIL” and “Stabl”, already used in to predict response to therapy in cancer disease and pre-term delivery, respectively), offering a new perspective on sepsis management in terms of outcome prediction.

Round 2

Reviewer 2 Report

Comments and Suggestions for Authors

The authors incorporated the changes suggested by the reviewers during the peer-review process. The manuscript can be accepted for publication in its current form.

Reviewer 3 Report

Comments and Suggestions for Authors

No additional comments to the authors.